# Synthetic Biology Toolkit for a New Species of *Pseudomonas* Promissory for Electricity Generation in Microbial Fuel Cells

**DOI:** 10.3390/microorganisms11082044

**Published:** 2023-08-09

**Authors:** Franciene Rabiço, Matheus Pedrino, Julia Pereira Narcizo, Adalgisa Rodrigues de Andrade, Valeria Reginatto, María-Eugenia Guazzaroni

**Affiliations:** 1Department of Biology, Faculty of Philosophy, Sciences and Letters of Ribeirão Preto, University of São Paulo, Ribeirão Preto 14040-901, Brazil; franciene.oliveira@usp.br (F.R.); matheus.pedrino.goncalves@usp.br (M.P.); 2Department of Chemistry, Faculty of Philosophy, Sciences and Letters of Ribeirão Preto, University of São Paulo, Ribeirão Preto 14049-900, Brazil; juliapnarcizoquimica@usp.br (J.P.N.); ardandra@usp.br (A.R.d.A.); valeriars@ffclrp.usp.br (V.R.)

**Keywords:** microbial fuel cells, *Pseudomonas*, pSEVA vectors, synthetic biology

## Abstract

Microbial fuel cells (MFCs) offer sustainable solutions for various biotechnological applications and are a crucial area of research in biotechnology. MFCs can effectively treat various refuse, such as wastewater and biodiesel waste by decomposing organic matter and generating electricity. Certain *Pseudomonas* species possess extracellular electron transfer (EET) pathways, enabling them to transfer electrons from organic compounds to the MFC’s anode. Moreover, *Pseudomonas* species can grow under low-oxygen conditions, which is advantageous considering that the electron transfer process in an MFC typically leads to reduced oxygen levels at the anode. This study focuses on evaluating MFCs inoculated with a new *Pseudomonas* species grown with 1 g.L^−1^ glycerol, a common byproduct of biodiesel production. *Pseudomonas* sp. BJa5 exhibited a maximum power density of 39 mW.m^−2^. Also, the observed voltammograms and genome analysis indicate the potential production of novel redox mediators by BJa5. Additionally, we investigated the bacterium’s potential as a synthetic biology non-model chassis. Through testing various genetic parts, including constitutive promoters, replication origins and cargos using pSEVA vectors as a scaffold, we assessed the bacterium’s suitability. Overall, our findings offer valuable insights into utilizing *Pseudomonas* spp. BJa5 as a novel chassis for MFCs. Synthetic biology approaches can further enhance the performance of this bacterium in MFCs, providing avenues for improvement.

## 1. Introduction

Bioelectrochemical systems (BESs) are an emerging sustainable technology for the simultaneous production of energy and wastewater treatment that has significantly increased during the past fifteen years [1,2]. Applications of BESs include microbial desalination cells (MDCs) for water desalting and microbial fuel cells (MFCs), which use microbial metabolism to generate energy from organic materials. Thus, microorganisms act as catalysts that provide a promising alternative for renewable energy generation [3,4,5,6,7]. Additionally, BESs present unique opportunities for the efficient and clean manufacture of high-value chemicals and fuels employing microorganisms, and, in addition, have great promise for the effective treatment of recalcitrant and harmful contaminants [8]. This process is called bioelectrochemical treatment (BET) or microbial electroremediation [9]. Due to its highly interdisciplinary approach, BESs have produced extensive research in various fields such as microbiology, electrochemistry, bioelectrochemistry, biotechnology, environmental science, and materials science. 

BESs systems work from the redox potentials of an oxidation reaction at the anode and a reduction reaction at the cathode providing a potential difference that acts as a catalyst for the spontaneous flow of electrons from a low to a high potential [10]. In addition, when an external voltage is introduced to that system, the electrons flow in the opposite direction from the way they did previously and, as a result, electricity is transformed into chemical energy [11,12]. Microorganisms are called bioanodes and biocathodes, respectively, depending on whether they catalyze anodic or cathodic processes [2,6]. The MFCs, which are the focus of our study, are a specific type of device that produces energy through the decomposition of organic material in the anode component. Bioelectricity is produced when organic material is oxidized at the anode at a low-redox potential and when oxygen is reduced at the cathode at a high-redox potential. The cathode potentials in microbial electrolysis cells (MECs) are subjected to chemical synthesis by the application of an external voltage [13].

MFCs can be classified into two different types, depending on how the electrons are transferred from the microorganisms to the anode. In MFCs that use synthetic mediators, electron carriers are added, while in mediator-free MFCs, no external mediators are added [14]. The exogenous synthetic mediators used to perform electron transfer are usually dyes [15,16,17]. However, toxicity and instability of the added mediators can limit their applications in MFCs [18]. In mediator-free MFCs, microorganisms employ electrochemically active cytochromes that facilitate electron transfer. Among these electrogenic microbes, we can mention *Geobacter* sp. [19] *Shewanella* sp. [20], *Clostridium* sp. [21], *Aeromonas* sp. [22] and *Pseudomonas* sp. [23] that can transfer electrons to extracellular electron acceptors such as oxidized metals and electrodes [24]. The extracellular electron transfer rate (EET) can be measured using MFCs, in which electrogenic microorganisms are grown as biofilm on the anode [25,26]. Some bacteria, such as *Pseudomonas aeruginosa,* are considered weak electrogens, meaning that they are capable of EET at a low rate [27]. In this case, EET is mediated by phenazines and pyoverdines [28].

*Pseudomonas* is a metabolically versatile genus that can be found in a variety of environments, including soil, water, human tissue and infected animals. As a result, it can grow on a variety of substrates in complex ecosystems and in oxygen-rich and anoxic environments [23]. Some *Pseudomonas* species, such as *P. aeruginosa*, are of interest due to their diverse metabolic activity, application in waste treatment generated in production and conversion of biodiesel, and their ability to produce compounds of importance such as antibiotics, siderophores and surfactants [29]. Their electrogenic properties are mainly due to phenazine-type molecules, which undergo reversible redox reactions [14]. Pyocyanin (PYO) is the most well-known phenazine from *P. aeruginosa* [30,31,32]. Its structure resembles pyoverdines, which are green fluorescent siderophore peptides widely found in the *Pseudomonas* genus. Pyoverdins are iron chelating molecules that supply this mineral to cells and whose predominant form occurs at neutral pH and in oxygenated environments [33]. While the need for oxygen may limit the use of these microbes in anoxic bioanodes, it was shown that alternating aerobic and anaerobic conditions can be sufficient to induce the production of these EET molecules and the growth of *P. aeruginosa* biofilm on the electrodes [34].

Due to the production of some natural mediator molecules and the metabolic ability of this microorganism to decompose pollutant oxidizable substrates and recover energy, *Pseudomonas* sp. are promising biocatalysts, being reported as one of the most representative genera in electrogenic biofilm consortia in MFCs in medium with acetate [35,36] or glycerol [37]. In addition to these natural characteristics, *Pseudomonas* can be easily modified with several synthetic engineering techniques and tools that can be useful to increase productivity in MFCs, such as increasing power density and current density [23,24,25,26,27,28]. Moreover, by synthetic biology techniques, EET kinetics can be improved, which is useful for microbial anodic electro-respiration and to promote pollutant removal at both the anode and cathode [1]. Additionally, it is possible to stimulate the growth of a biofilm on the electrode, increasing the number of microorganisms that can be used for subsequent catalytic reactions [38]. Moreover, in the context of pollutant decomposition, synthetic biology approaches offer the potential to enhance and regulate the production of enzymes in bacteria that are involved in the breakdown of various pollutants [39].

Here, we evaluated the aptness of a new electrogenic and environmental strain of *Pseudomonas* sp. named BJa5. In this work, it was seen that *Pseudomonas* sp. BJa5 has a promising performance in MFCs, so it is important to study the suitability of this strain to be used as a synthetic biology chassis under usual laboratory conditions. Thus, establishing the experimental protocols for the successful use of molecular tools in BJa5 would allow genetically modifying this bacterium, enabling the enhancement of BJa5 function in MFCs. For example, BJa5 could be used as a carrier chassis for synthetic circuits to enhance its external electron transfer capacity by adding or modulating the expression of mediator molecule synthesis operons, such as phenazines and pyoverdines. In the same way, BJa5 could be genetically modified to be more resistant to low oxygen environments and to the presence of toxic chemical compounds by adding genes that confer resistance to biomass hydrolysate inhibitors.

## 2. Materials and Methods

### 2.1. Bacteria Strains, Synthetic Biology Tools and Growth Conditions

The *Pseudomonas* sp. BJa5 strain was prospected from soil localized in Ribeirão Preto, Brazil (21°10′40″ S; 47°48′36″ W). *Escherichia coli* DH5α and *Pseudomonas* spp. strains were grown aerobically at 200 rpm and 25 °C, in M9 medium (47.7 mM Na_2_HPO_4_, 22 mM KH_2_PO_4_, 8.6 mM NaCl_2_, 18.7 mM NH_4_Cl, 2 mM, MgSO_4_, 0.1 mM CaCl_2_) or in Luria-Bertani (LB) medium, according to the method used in each assay. As appropriate, 50 μg/mL Kanamycin (Km) was added to the media. Bacterial strains and plasmids used in this study are detailed in Table 1.

### 2.2. Microbial Fuel Cell Design and Operation

The L-shaped, double-chamber MFC was made of glass with an anode compartment volume of 35 mL, containing the working electrode, which consisted of a 9 cm^2^ carbon cloth suspended by a platinum wire. The 16 cm^2^ cathode (gas-diffusing layer-type carbon cloth, 0.3 mg cm^−2^ PtC 40%) was hot-pressed at 130 °C and 35 kgf cm^−2^ for 180 s with a proton exchanger membrane (Nafion 117), creating an air-breathing cathode. The external circuit was connected to a 2200 Ω resistor and an Arduino Mega 2560 (Arduino, Turin, Italy) microcontroller board (C++ language) coupled to a computer for the acquisition of voltage. There was no control of O_2(g)_ diffusion in the anodic chamber. The culture medium employed was described by [44], containing in (g.L^−1^): NaHCO_3_ 2.5, Na_2_HPO_4_ 0.6, NaH_2_PO_4_.H_2_O 0.74, CaCl_2_.2H_2_O 0.1, KCl 0.1, NH_4_Cl 1.5, NaCl 0.1, MgCl_2_.6H_2_O 0.1, MgSO_4_.7H_2_O 0.1, MnCl_2_.4H_2_O 0.005, Na_2_Mo_4_.2H_2_O 0.001, yeast extract 0.05 and glycerol at 1.0 as carbon source. 

### 2.3. Electrochemical Measurements

Cyclic voltammetry (CV) was recorded using AUTOLAB PGSTAT 30 (Metrohm/Eco Chemie, Herisau, Switzerland) potentiostat controlled with NOVA 1.11 software. Analyzes were conducted on the MFC in the presence of an Ag/AgCl reference electrode. The scan speed was set at 20 mV.s^−1^, and the investigated potential window ranged from −0.6 to 0.6 V.

### 2.4. Coulombic Efficiency

The Coulombic efficiency (CE) was defined as the ratio of actual charge, which was produced in MFC, to the theoretical possible Coulombs from glycerol. The actual charge or Coulombs (C_p_) was determined by integrating the MFC current over time. The theoretical Coulombs (C_Ti_) that can be produced from glycerol was calculated as [45],
(1)$$C_{\mathrm{Ti}}=\frac{{\mathrm{Fb}}_{i}S_{i\;}v}{M_{i}}$$
where F is Faraday’s constant (98.485 C/mol of electrons), b_i_ is the number of mol of electrons produced per mol of glycerol (14), S_i_ (g/L) is the consumed substrate concentration, v (L) the anode compartment liquid volume, and M_i_ the molecular weight of the substrate (92.09 g/mol).

So, the Coulombic efficiency for the MFC was calculated as follows,
$$\mathrm{CE}=\frac{\mathrm{Cp}}{C_{\mathrm{Ti}}}\times 100\%$$

### 2.5. Transformation of Pseudomonas sp. Strains

Electrocompetent *Pseudomonas* sp. BJa5 and *P. putida* KT2440 were prepared after three washes with a solution of 20 mM sucrose. Each strain was cultivated in LB at 30 °C for overnight (16 h). Then, 10 mL of each culture was centrifuged at 16,000 rpm 1 min at 4 °C; cells were washed three times with 5 mL with a cold solution of 20 mM glucose and resuspended in 400 μL of the same solution on ice. All processes were realized at a chill temperature and the electroporation occurred immediately. A 50 μL volume of electrocompetent cells was transformed with 2 ng of plasmid in a MicroPulser Electroporator (BioRad, Sao Paulo, Brazil) using program Ec1 (V = 1.8 kV) and 0.1 mm cuvettes. After electroporation, 950 μL of LB was added and mixed with the freshly transformed cells. Next, cells were incubated for 1 h at 30 °C. Seventy microliters were plated in a LBKm.

### 2.6. Growth Curves

*Pseudomonas* sp. BJa5 and *P. putida* KT2440 were characterized under four carbon sources to compare their metabolic abilities. Single colonies of each strain were cultivated in M9 medium supplemented with sole carbon sources (0.2% acetate, 0.3% benzoate, 1% citrate and 1% glycerol) at 30 °C and 200 rpm for 16 h. Then, each inoculum was diluted at OD_600nm_ ~ 0.05 in a 12-well plate with M9 medium 0.2% acetate, 0.3% benzoate, 1% citrate and 1% glycerol and incubated at 30 °C in a VICTOR^®^ Nivo^TM^ Multimode Microplate Reader (Perkin-Elmer, Waltham, MA, USA) for 24 h. OD_600nm_ measurements were taken at each 2 h. The growth curves were constructed using three biological replicates for each strain. The specific growth rates (μ) were calculated for each strain in all conditions tested using the linear method [46,47].

### 2.7. Plasmid Replication, and Structural and Segregational Stability

In order to design a suitable vector for a broad range of bacterial hosts, the origin of replication must be recognized by them. For this purpose, we evaluated the suitability of origins RK2, pBBR1 and pRO1600 derived from the Standard European Vector Architecture (SEVA) [48] (http://seva.cnb.csic.es/, accessed on 20 March 2023). The pSEVA plasmids are modular vectors that contain three exchangeable modules: replication origin, selection marker, and cargo. For all of the constructs used in this work, the selection marker was Km and the replication origins were as follows: RK2 origin (plasmid pSEVA228), pBBR1 origin (plasmid pSEVA231 and pSEVA232) and pRO1600 origin (plasmid pSEVA247Y). All plasmids were introduced in *Pseudomonas* spp. by electroporation, and transformation efficiency was determined for each origin of replication. 

To determine segregational stability, all the strains were cultivated in LBKm at 30 °C for 16 h, diluted at OD_600nm_ ~ 0.1 in LB without antibiotics, and cultivated at 30 °C for 2, 4, 7 and 24 h. The diluted strains were plated in LB agar at each point. After growth, 12 colonies containing each replication origin for each time were selected from LB plates, then plated on LB with and without Km. Then, the percentage of cells retaining Km resistance was determined. The experiment was performed with three independent replicates using three biological replicates.

The structural stability of plasmids was evaluated from those 12 colonies plated in LBKm at 24 h. For this, three colonies were selected for each construct, and cultivated in LBKm at 30 °C for 16 h. Next, plasmids were extracted, digested with *PacI* and *SpeI* restriction enzymes (New England Biolabs Inc., Ipswich, MA, USA), and the restriction patterns were analyzed by electrophoresis on agarose gels.

### 2.8. Fluorescence Protein (mCherry and GFPlva) Expression in BJa5

For evaluation, the fluorescence protein expression used the vector pVANT and pSEVA231 harboring the mCherry and GFPlva protein, respectively, as previously described by Siqueira, Guazzaroni and Amarelle et al. [42,43]. The pVANT vector was designed using the pSEVA231 with Km as selection marker and pBBR1 replication origin. In order to determine whether *Pseudomonas* sp. BJa5 recognizes canonical constitutive promoters, we tested BBa_J23100 (P*_j_*_100_, a high-strength promoter), BBa_J23106 (P*_j_*_106_, a medium-strength promoter with 50% activity compared to BBa_J23100), and BBa_J23114 (P*_j_*_114_, a low-strength promoter with 10% activity compared to BBa_J23100) sequences of *E. coli* (http://parts.igem.org/Promoters/Catalog/Anderson, accessed on 20 March 2023). To construct the reporter vectors, the mCherry gene was cloned between the *HindIII* and *SpeI* restriction sites in the pVANT vector and the GFP gene was cloned between the *EcoRI* and *HindIII* in the pSEVA231 [43]. After this, the promoters’ sequence was introduced into the plasmid via Gibson assembly. A promoterless vector was used as a negative control (referred as “no pr”). All of the plasmids were introduced by electroporation into *Pseudomonas* sp. BJa5 and *P. putida* KT2440.

#### 2.8.1. Promoter Activity as a Function of Genomic Context

Bacterial strains were grown in LBKm at 30 °C for 16 h. All cultures were normalized to an OD_600nm_ ~ 1, and 20 μL drops were spotted in LBKm plates. Cultures were incubated at 30 °C, and promoter activity was observed on spots by red fluorescent protein, derived from DsRed (mCherry). The emission was detected using the Safe Imager 2.0 blue-light transilluminator (Invitrogen, Carlsbad, CA, USA). The experiment was repeated three times with different biological replicates.

#### 2.8.2. Quantification of Promoter Activity

Strains were grown in M9 medium with 1% citrate at 30 °C for 16 h. Next, all cultures were normalized to an OD_600nm_ ~ 0.05 and incubated at 30 °C in a VICTOR^®^ Nivo^TM^ Multimode Microplate Reader (Perkin-Elmer) for 6 h. mCherry fluorescence was detected using a 580 nm excitation filter/625 nm emission filter with D500 optical mirrors. Quantitative relative fluorescence was expressed as fluorescence emission normalized by OD (625 nm/OD_600nm_). Fluorescence analysis was performed using three biological replicates.

### 2.9. Heterologous Expression of an Endoglucanase Cel5A

The plasmid containing the endoglucanase 5A (Cel5A) from *Bacillus subtilis* 168 coding sequence was previously constructed by [40] in pSEVA232 as a cloning vector. pSEVA232-*Cel5A* was introduced in *Pseudomonas* sp. strains by electroporation and transformants were selected in LBKm plates. An empty vector was used as negative control. To determine the activity of the Cel5A enzyme, a qualitative assay was performed with carboxymethyl cellulose (CCM) and trypan blue. In this assay, *Pseudomonas* sp. BJa5 and *P. putida* KT2440 were cultivated in LBKm at 30 °C for 16 h; cell cultures were normalized at OD_600nm_ ~ 1. Drops of 10 μL were placed in LBKm plates, supplemented with 2.5% (*w*/*v*) carboxymethyl cellulose and 0.01% (*w*/*v*) trypan blue solution [49]. Plates were incubated at 30 °C and Cel5A activity was evidenced as a halo formed around the drops after 48 h. *E. coli* DH5*α* strain expressing Cel5A was used as a positive control. The halos of each strain with the Cel5A plasmid were measured and represented in centimeters. The experiment was repeated three times with different biological replicates.

### 2.10. Genome Sequencing

Total DNA of *Pseudomonas* sp. BJa5 strain was extracted using the GeneJET Genomic Purification Kit (Thermo Fischer Scientific, Waltham, MA, USA), according to the manufacturer’s instructions. The concentration of DNA was determined fluorometrically using the Qubit^®^ 3.0 (Qubit^®^ dsDNA Broad Range Assay Kit, Life Technologies, Carlsbad, CA, USA). The DNA library was prepared using the Nextera XT DNA Library Prep Kit (Illumina, San Diego, CA, USA), assessed for quality using the 2100 Bioanalyzer (Agilent Genomics, Santa Clara, CA, USA), and subsequently submitted to sequencing using the Illumina HiSeq (2 × 150 bp) platform (Illumina, San Diego, CA, USA). Paired-end reads were trimmed and assembled using SPAdes v.3.15.4 [50,51]. The completeness, quality, and contamination were assessed using Type Strain Genome Server (TYGS) [52]. Draft genomes were annotated using Prokka v.1.14.5. The *Pseudomonas* sp. BJa5 genome is available at NCBI under GenBank accession number JALRNG000000000.1.

The identification of electrogenic-associated genes (*phzA1-B1*, *phzA2-B2*, and *pvd*) was first performed by gene annotation followed by BLASTn. Subsequently, the gene sequences were extracted, aligned, and compared to reference strain *P. aeruginosa* PAO1 genome (GenBank accession number AE004091) using Geneious Prime^®^ v. 2023.1.2 (Biomatters Ltd. Auckland, New Zealand). 

### 2.11. Taxonomic Affiliation of Pseudomonas sp. BJa5

The phylogenetic analysis of *Pseudomonas* sp. BJa5 strain was performed using the whole-genome sequencing data. The phylogenetic identification of the strain was performed using the TYGS platform and the classification was based on Digital DNA-DNA hybridization (dDDH) values [52]. Bootstrap analyses (1000 replicates) were performed to assess the reliability of the inferred branches.

## 3. Results and Discussion

### 3.1. Production of Bioelectricity and Electrochemical Activity of Strain BJa5 Is Mediated by Potential New Redox Molecules

In order to know *Pseudomonas* sp. BJa5 suitability to transfer electrons from organic compounds to the anode of an MFC, we tested the BJa5 strain as a biocatalyst in MFC fed with glycerol. We monitored the production of electric current in the MFC, with an external resistance of 2200 Ω, during two feeding cycles. In the absence of bacteria, no current was recorded in the MFC. After 24 h from inoculation on the anodic compartment with the strain BJa5, electricity could be recorded (Figure 1). This period might be associated with bacterial growth and the concentration of redox mediators that help in the external transfer of electrons. The average electric current and CE, based on electron recovery from consumed glycerol, during a stable operation cycle of the MFC were, 38.8 ± 5.8 mA.m^−2^ and 0.6%, respectively.

Several factors affect the performance of an MFC, particularly the electronic transfer mechanism from the biocatalysts to the electrode, substrate type, charge and reactor design [53]. *P. aeruginosa* is the representative species of the genus *Pseudomonas* as a biocatalyst in MFCs. For example, Ali et al. (2017) [54] achieved 0.26 mA.m^−2^ with a *P. aeruginosa* strain from a domestic sewage using sucrose as carbon source. In another study, Yong et al. (2017) [34] obtained 23 mA.m^−2^ by testing the strain *P. aeruginosa* CGMCC1860 with glucose. Thus, glycerol oxidation performed by the BJa5 strain demonstrated promising results compared to those reported for *P. aeruginosa*, using sugar as substrate. Moreover, glycerol is the main constituent of the most abundant residue generated by biodiesel production [55]. Thus, the possibility to oxidize glycerol in MFC can afford energy generation from a residue, making this an environmentally friendly approach.

The CV of the anode can shed some light into the electrochemical activity and the electron transfer mechanism of the bacteria to the electrode. Thus, the CV assay was performed in the anodic compartment under two different conditions (Figure 2): first, the anode was clean, without the biofilm, and only the planktonic cells and their metabolites were left in the anodic compartment; and the second condition was with the anode with biofilm in a fresh and sterile medium. In the presence of planktonic cells (Figure 2, red line), the oxidation and reduction peaks were observed at −0.070 and −0.24 V (vs. Ag/AgCl), respectively. When biofilm was used as an anode (Figure 2, blue line), the oxidation peak is less evident, however, occurs at the same potential region observed with the planktonic cells, suggesting a similar electrochemical signal, e.g., electrons are transferred by mediated electron transfer (MET). In MET, redox-active molecules that act as mediators of electron transfer are reduced on the external surface of the cell and oxidized when donating electrons to the electrode [56]. A considerable number of *Pseudomonas* species produce phenazine derivatives, such as PYO and phenazine-1-carboxamide (PCA) [57], which act as electron carriers [34]. In addition, pyoverdins, a class of yellowish-green siderophores, also exhibit redox properties and were reported as EET mediators in MFC [28]. Unfortunately, the redox signals observed in the voltammograms and the analysis of the BJa5 genome do not suggest that phenazine and pyoverdin are the redox-active species produced by BJa5. 

Moreover, we could not identify either the phenazine biosynthesis genes (*phzA1-A2*, *phzB1-B2*) and the complete pyoverdine region (*pvd* locus), which are intimately associated with MET in *Pseudomonas* genus. On the other hand, we identified the *pvdA* and *pvdQ* genes involved, respectively, with chromophore incorporation in siderophores and maturation [58]. PvdA protein acts in cytoplasm giving the basis of PYO incorporating chromophores or other moieties [31]. On the other hand, PvdQ protein is localized in periplasm and participates in biochemical reactions involved in maturation of PYO to a fluorescent-form, culminating in ferribactin production [59]. As we could not identify the *pvd* genes that act directly in PYO chain biosynthesis, we suggest that pvdA and pvdQ might be accessory genes to produce new redox mediators. Due to broad activity, these genes might act in (i) secondary reactions that generate essential precursors or in (ii) maturation of molecules to produce new redox fluorescent mediators that are not directly associated with classical MET and PYO production. Notably, future efforts must be employed in identifying and elucidating the molecular mechanisms of these potential new redox molecules. 

### 3.2. BJa5 Strain Is a Potential New Pseudomonas Species

Considering the bioelectrochemical potential of strain BJa5 described in MCFs, we decided to obtain more information about this bacterium using genomic approaches. *Pseudomonas* sp. BJa5 is an environmental isolate obtained from soil samples in the region of Ribeirão Preto, Brazil. The final draft assembly of the genome contained 425 contigs with a mapping coverage of 300×. Also, the genome contains 4978 coding sequences and 71 RNAs, of which 67 were tRNAs, 3 were rRNAs and 1 was tmRNA.

Based on TYGS analysis and digital DNA–DNA hybridization (dDDH) values, we confirmed that BJa5 belongs to the genus *Pseudomonas* (Figure 3). The *Pseudomonas* sp. BJa5 is in the same monophyletic group as *Pseudomonas vranovensis* and closely related to the *Pseudomonas alkylphenolica* and *Pseudomonas hutmensis* species with a 100% bootstrap. However, BJa5 genome has a maximum dDDH_4_ value around 30% when compared with other *Pseudomonas* sp. genomes (Figure 3). Thus, according to dDDH4 genomic metrics, *Pseudomonas* sp. BJa5 is a potential new *Pseudomonas* species [60]. 

The size and GC content of BJa5 genome was estimated to be, respectively, 5,650,681 Mb and 63.56% (Figure 3). It is considered that both GC content and genome size are important taxonomic markers, especially for genus. Genome size differences vary within genus, as does the corresponding gene content, and in this case, genome size is directly related to the number of coding sequences (CDS). According to the current literature, in *Pseudomonas* genus, GC content ranges from 48.3% to 68.3% [61,62] and coding genes range from 2803 (genome size 3.03 Mb) to 6895 (genome size 7.38 Mb) [63]. Therefore, both the genome size, CDS number, and GC content of the BJa5 genome are within the expected ranges for bacteria belonging to the genus *Pseudomonas.*

**Figure 3 microorganisms-11-02044-f003:**
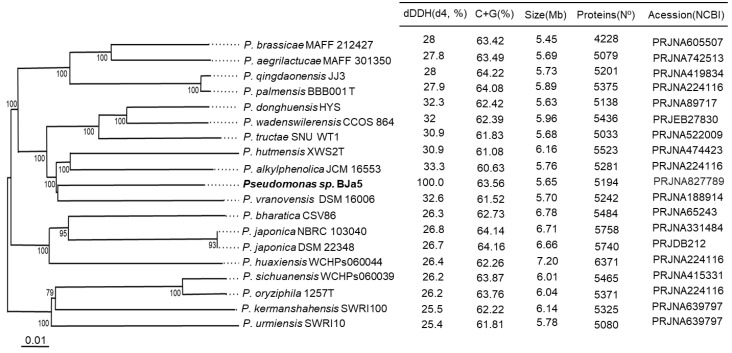
Taxonomic affiliation of *Pseudomonas* sp. strain BJa5 (highlighted in bold letter). Dendrogram of BJa5 and related species inferred with FastME [64] from Genome BLAST Distance Phylogeny method (GBDP) distances calculated from genome sequences. Branch lengths are scaled in terms of GBDP distance formula d5; numbers above branches are GBDP pseudo-bootstrap support values from 100 replications. The dendrograms were constructed using the TYGS tool.

### 3.3. BJa5 Can Grow on Different Carbon Sources of Organic Acids

Free-living bacteria have a versatile metabolism that facilitates many compounds with carbon in their structure, which can be used as an energy source. This ability facilitates survival in different habitats and adaptation to changing environmental conditions. An example of these bacteria is *Pseudomonas*, which are ubiquitous in various environmental conditions [65]. A common characteristic of all of them is their considerable metabolic versatility, being able to assimilate a wide range of compounds. The preferred carbon sources of *Pseudomonas* are organic acids or amino acids [66], and there is a sequential ranking in which some organic acids and amino acids are the preferred consumed compounds and hydrocarbons are the least preferred compounds, such as glucose. The hierarchy of substrate preferences for *Pseudomonas* are generally succinate, followed by citrate, lactate, acetate, some amino acids, and finally glucose [66,67]. As organic acids are typically found in waste residues, its utilization by *Pseudomonas* species aligns with eco-friendly processes by promoting waste management, renewable energy production, and reduced environmental impact. Furthermore, organic acids are a cheap and highly available carbon source for industrial bioprocesses.

Based on this, we tested four different carbon sources (acetate, benzoate, citrate, and glycerol) that were used to identify the most suitable condition for *Pseudomonas* sp. BJa5 growth in comparison with the well-known reference strain *P. putida* KT2440 [42]. The fastest growth of each strain was observed when using the M9 medium with 1% citrate (Figure 4A,B). In both strains, we observed a rapid exponential growth that started after a short-lag phase of approximately 2 h for KT2440 and 3 h for BJa5.

*Pseudomonas* sp. BJa5 shows similar growth to *P. putida* KT2440 when both are cultivated in medium containing citrate as carbon source. Citric acid is a ubiquitous compound in nature, being present in fruits and in human tissues such as teeth, and acts as a key cellular intermediate [68]. In *P. putida* and *P. aeruginosa*, citrate enters carbon metabolism directly into the tricarboxylic acid (TCA) cycle through the gluconeogenic pathway [69,70]. Thus, as seen in other *Pseudomonas*, citrate is well assimilated and utilized as a carbon source in BJa5. 

Also, the other carbon sources tested were able to maintain the growth of both strains. However, BJa5 strain showed a higher growth on the medium with 0.2% acetate and a similar growth as strain KT2440 on the medium containing 0.3% sodium benzoate and 1% glycerol (Figure 4A,B). These differences in growth can be better observed when we compare the growth rates of both strains (Figure 4C). Therefore, we stated that the BJa5 strain is a new potential species able to grow on a variety of carbon sources that are cheap and abundant, with no restriction in this regard.

An interesting finding about BJa5 strain is that it obtained the highest growth rate when fed with acetate: a microbial growth inhibitor. In the tested conditions of this study, a concentration of 2 g/L of acetate was used, however, in the literature, it was shown that a contraction greater than 5 g/L can inhibit cell growth [67,71]. However, acetate is a promising substrate due to its economic advantage [72] and sustainability, as it is present in lignocellulosic biomass hydrolysates [73]. Furthermore, acetate was used as a substrate for the production of polyhydroxyalkanoates (PHA) [74]. *P. putida* KT2440, although able to produce PHA [75], has limited and slow growth when acetate is used as the unique carbon source [76]. Thus, the insufficient assimilation of this compound by *P. putida* KT2440 hinders the development of this strain as an efficient microbial cell factory [77]. Therefore, *Pseudomonas* sp. BJa5 might be interesting as a future model to produce PHA from acetate.

### 3.4. Modular pSEVA Plasmids Are Appropriate Vectors for Pseudomonas sp. BJa5

To use a non-model chassis successfully in biotechnological applications, tools and protocols customized to the specific microorganism must be developed. One of the most important steps is finding suitable replication origins. Here, we tested three different replication origins in the *Pseudomonas* sp. BJa5 strain: RK2, pBBR1 and pRO1600. When transformed with each of these replication origins, the BJa5 strain obtained high-efficiency levels (approximately 10^7^−10^8^ CFU/μg DNA, Appendix A).

Besides the requirement of being capable of introduction and replication within the host cell, a vector must also possess the ability to maintain its intact structure without integrating into the bacterial genome for multiple generations. Considering this, the structural and segregational stability of plasmids with different replication origins was evaluated. Resistance to the antibiotic kanamycin was maintained for all clones and plasmids analyzed in the BJa5 strain (Appendix A) and also in the KT2440 strain. Accordingly, plasmids were successfully maintained extrachromosomally without undergoing structural changes in both strains, as confirmed through plasmid extraction and restriction analysis by *PacI* and *SpeI* enzymes (Appendix A).

Hence, in both tested bacterial strains, the pSEVA plasmids carrying various origins exhibited notably high-transformation efficiencies and consistently preserved their structural integrity across successive generations. This versatility and ability to recognize diverse replication origins represent significant progress towards utilizing BJa5 strain in BESs. Given these characteristics, the pSEVA plasmids emerge as an excellent vector choice for *Pseudomonas* sp. BJa5.

### 3.5. Pseudomonas sp. BJa5 Recognizes the P_j100_ Canonical Promoter

Promoter sequences are also important tools for synthetic biology considering that the expression of a heterologous gene depends on the recognition of these regions by the host transcriptional machinery. In this work, we tested the canonical constitutive promoter sequence BBBa_J23100 (P*_j_*_100_), which is considered a strong promoter in *E. coli* [78] and was shown to be functional also in other environmental *Pseudomonas* spp. strains [42,79]. Thus, promoter P*_j_*_100_ was cloned in the pVANT vector to control expression of the mCherry reporter gene and tested in *P. putida* KT2440 and *Pseudomonas* sp. BJa5. As shown in Figure 5A, the P*_j_*_100_ promoter sequence is functional in both strains and it is possible to visualize the fluorescence of the mCherry protein after incubation at 30 °C for 16 h. However, reporter gene expression is lower in the BJa5 strain compared to *P. putida* KT2440 (Figure 5B). In addition, expression of medium (P*_j_*_106_) and low (P*_j_*_114_) constitutive promoters with the GFP reporter gene was tested in both strains, although no fluorescent signal was observed in the BJa5 strain for both promoters (Appendix A). In the case of *P. putida* KT2440, the fluorescent signal was weakly observed only with the medium strength promoter P*j*_106_, similar to that obtained by Amarelle et al. (2023) [42]. Considering these results, we can conclude that the BJa5 strain can efficiently recognize the P*_j100_* promoter, thus allowing the expression of synthetic circuits in this host. Nevertheless, we observed that the fluorescence signals were not consistently stable during extended periods of passage culture in BJa5. Consequently, standardization of current protocols is still necessary to identify the optimal conditions for this assay.

### 3.6. Pseudomonas sp. BJa5 Strain Expresses the Heterologous Enzyme Cel5A

An experiment was carried out as a proof of concept to verify whether expression and functionality of heterologous genes occurs in the strain *Pseudomonas* sp. BJa5. For this, a plasmid containing an endoglucanase (pSEVA232-Cel5A) was introduced into *P. putida* KT2440 and *Pseudomonas* sp. BJa5. As shown in Figure 6A, both *Pseudomonas* strains were able to express Cel5A after 48 h of incubation. The halos formed in each strain were measured relative to Cel5A expression. The DH5α strain exhibited the highest magnitude of Cel5A expression, followed by BJa5 and KT2440 in decreasing order (Figure 6B). Thus, the BJa5 strain was able to express a functional Cel5A enzyme under usual conditions (30 °C), in comparison with the reference bacterium *P. putida* KT2440. 

As expected, *E. coli* is an ideal host for recombinant protein expression, demonstrated by higher Cel5A activity, which might be attributed to well-characterized transcriptional machinery, and an ability to render higher levels of protein production [80]. However, microbial hosts with new abilities, especially related to secretion, are promising candidates for recombinant expression such as *Pseudomonas* spp. For example, cellulases expressed in *P. putida* KT2440 showed similar bioactivity compared with *E. coli*, even for a wide range of temperatures and pH [81]. In this context, the BJa5 strain was able to express a functional Cel5A enzyme under usual conditions (30 °C) in comparison with the reference bacterium *P. putida* KT2440: a trait that could be explored in future efforts to address its functionality in recombinant protein expression.

## 4. Conclusions

The discovery of novel bacteria capable of serving as chassis in bioelectrochemical systems holds significant potential. As an emerging field, the quest for new bacteria to enhance and optimize these systems is crucial. In addition to demonstrating the bioelectricity production of the novel *Pseudomonas* sp. BJa5 through glycerol oxidation, which shows promising results compared to sugar-based substrates commonly used by *P. aeruginosa* [54,55], our study highlights the potential of glycerol oxidation in microbial fuel cells (MFCs) as an environmentally friendly approach for energy generation from residues. Moreover, the observed voltammograms and BJa5 genome analysis indicate the potential production of novel redox mediators by BJa5, distinct from the phenazine and pyoverdin classes commonly found in the *Pseudomonas* genus.

In addition, synthetic biology tools can be used to modify existing biological systems to perform specific functions [82]. This includes the selection or modification of *microorganisms* to increase their ability to transfer electrons, improve the efficiency of bioelectrochemical reactions, and the production of specific compounds that can be used as substrates for energy generation. As an example, synthetic biology is being used to perform faster processes using EET pathways by refactoring split ferredoxins as switchable electron gates [83]. In summary, the results obtained in this work proved that *Pseudomonas* sp. BJa5 is a suitable new chassis that can be implemented in MCFs using pSEVA plasmids. The tested toolbox includes three different replication origins (RK2, pBBR1 and pRO1600), a selection marker (Km), two cargos, and a canonical constitutive promoter (P*_j100_*). In addition to these toolboxes, we successfully tested the expression of the reporter gene mCherry and also the enzyme Cel5A in the two *Pseudomonas* assayed. In the future, these tools can be further developed and utilized to design synthetic genetic circuits aimed at enhancing energy generation efficiency in *Pseudomonas* sp. BJa5. This will enable significant improvements in the performance of this bacterium within MFCs.

## Figures and Tables

**Figure 1 microorganisms-11-02044-f001:**
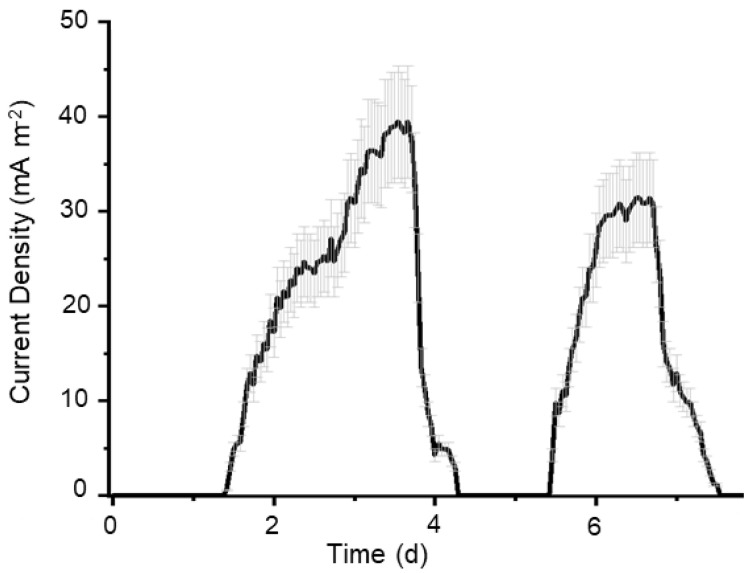
Production of electric current in MFC from strain *Pseudomonas* sp. BJa5. Current density (black line) produced in the MFC with an external resistor at 2200 Ω, inoculated with BJa5 strain and fed with glycerol at 1 g.L^−1^. The grey lines represented the current density deviation of the replicates.

**Figure 2 microorganisms-11-02044-f002:**
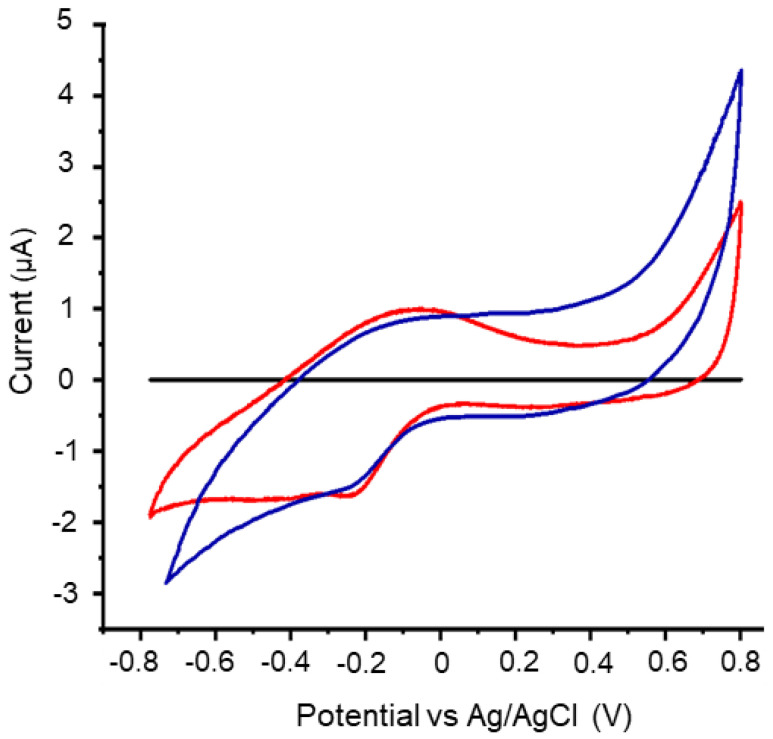
Cyclic voltammogram of the anode with planktonic cells (red line) and of the anode with biofilm in fresh and sterile medium (blue line). Abiotic control is represented by black line. ν = 20 mV.s^−1^, pH 6.9 (vs. Ag/AgCl).

**Figure 4 microorganisms-11-02044-f004:**
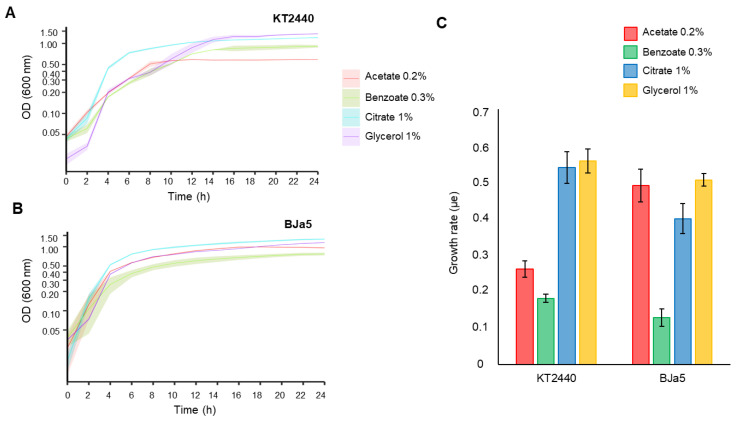
*Pseudomonas* sp. strain BJa5 (**A**) and *P. putida* KT2440 (**B**) growth curves into different carbon sources. The growth curves were constructed using three biological replicates. The graph represents the average OD_600nm_ from three replicates and SD (smoky in lines). A 3.0 factor should be applied to correct optical path lengths for panel (**A**,**B**). (**C**) Growth rates were determined at exponential phase for each carbon source using the linear method [46,47] represented by average ± SD.

**Figure 5 microorganisms-11-02044-f005:**
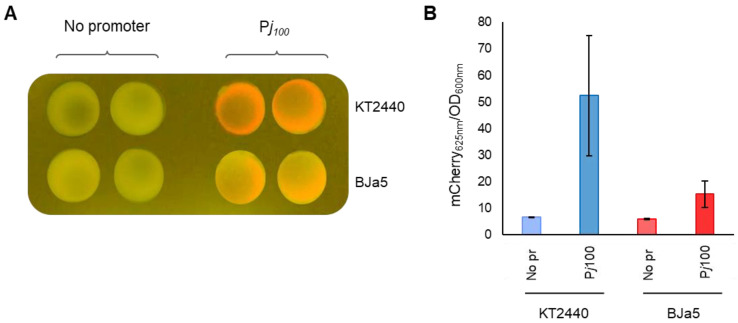
Activity of the canonical P*_j100_* promoter at 30 °C. (**A**) Functionality of the canonical BBa_J23100 (P*_j_*_100_) promoter was evaluated at 30 °C in *P. putida* KT2440 and in *Pseudomonas* sp. BJa5 using the mCherry reporter gene and the pVANT empty vector. The plasmid containing mCherry without a promoter (no pr) was used as a negative control. The assay was performed three times and similar results were obtained. (**B**) Expression of the canonical promoters at 30 °C was quantified at 16 h. Fluorescence was normalized to OD600nm. Data are the mean of three biological replicates, each with technical triplicates ± SD.

**Figure 6 microorganisms-11-02044-f006:**
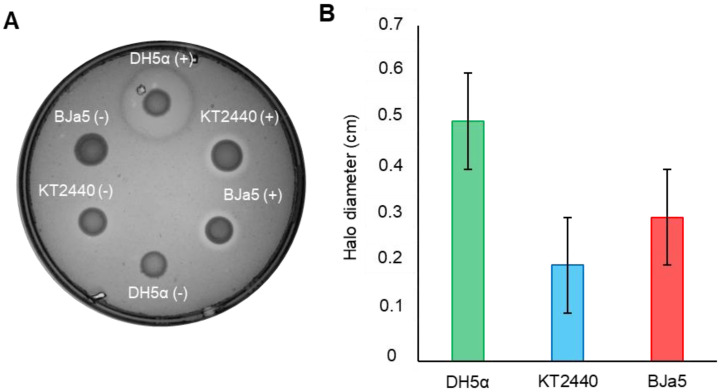
Activity of endoglucanase 5A (Cel5A) enzyme at 30 °C. (**A**) The functionality of the Cel5A enzyme was evaluated at 30 °C in *P. putida* KT2440 and BJa5 using the canonical promoter Bba_J23100 (P*_j100_*) and the pSEVA232 vector. The results were compared with *E. coli* DH5*α*. The plasmid containing Cel5A without a promoter (−) was used as a negative control. The presence of halos on the Petri dishes indicates the expression and activity of the enzyme. The assay was performed three times and similar results were obtained. (**B**) Measurements, in centimeters, of the halos formed by Cel5A activity, considering the relative size of the droplet bacterial culture. Data are the average of three biological replicates, each with technical triplicates ± SD.

**Table 1 microorganisms-11-02044-t001:** Features of wild-type and transformed bacterial strains.

Bacterium	Characteristics	Reference
DH5*α*	F–φ80lacZΔM15 Δ(lacZYA-argF)U169 recA1 endA1 hsdR17(rK–, mK+) phoA supE44 λ–thi-1 gyrA96 relA1	[40]
DH5*α* pSEVA232	DH5*α* carrying plasmid pSEVA232 (Km^R^)	[40]
DH5*α* pSEVA232_Cel5A	DH5*α* carrying plasmid pSEVA232_Cel5A*,* and endoglucanase from *Bacillus subtilis* 168 (Km^R^)	[40]
*Pseudomonas putida* KT2440	Reference bacteria	[41]
KT2440 pSEVA 228	KT2440 carrying plasmid pSEVA228 (Km^R^)	[42]
KT2440 pSEVA 232	KT2440 carrying plasmid pSEVA232 (Km^R^)	[42]
KT2440 pSEVA 247y	KT2440 carrying plasmid pSEVA247Y (Km^R^)	[42]
KT2440 pVANT	KT2440 carrying plasmid pVANT (Km^R^)	[43]
KT2440 pVANT-*Pj_100_mCherry*	KT2440 carrying plasmid Pvant-*Pj100*-*mCherry* (Km^R^)	This work
KT2440 pSEVA231	KT2440 carrying plasmid pSEVA231 (Km^R^)	[42]
KT2440 pSEVA231-*Pj_100_GFP*	KT2440 carrying plasmid PSEVA231-*Pj100*-*GFP* (Km^R^)	[42]
KT2440 pSEVA231-*Pj_106_GFP*	KT2440 carrying plasmid PSEVA231-*Pj106*-*GFP* (Km^R^)	[42]
KT2440 pSEVA231-*Pj_114_GFP*	KT2440 carrying plasmid PSEVA231-*Pj114*-*GFP* (Km^R^)	[42]
KT2440 pSEVA232_Cel5A	KT2440 carrying plasmid Pseva232-Cel5A (Km^R^)	[40]
*Pseudomonas* sp. BJa5	Bacterium isolated from garden soil in Ribeirão Preto, Brazil	This work
BJa5 pSEVA 228	BJa5 carrying plasmid pSEVA228 (Km^R^)	This work
BJa5 pSEVA 232	BJa5 carrying plasmid pSEVA232 (Km^R^)	This work
BJa5 pSEVA 247y	BJa5 carrying plasmid pSEVA247Y (Km^R^)	This work
BJa5 pVANT	BJa5 carrying plasmid pVANT (Km^R^)	This work
BJa5 pVANT-*P_j100_mCherry*	BJa5 carrying plasmid Pvant-*Pj100 mCherry* (Km^R^)	This work
BJa5 pSEVA231	BJa5 carrying plasmid PSEVA231 (Km^R^)	This work
BJa5 pSEVA231-*Pj_100_GFP*	BJa5 carrying plasmid PSEVA231-*Pj100*-*GFP* (Km^R^)	This work
BJa5 pSEVA231-*Pj_106_GFP*	BJa5 carrying plasmid PSEVA231-*Pj106*-*GFP* (Km^R^)	This work
BJa5 pSEVA231-*Pj_114_GFP*	BJa5 carrying plasmid PSEVA231-*Pj114*-*GFP* (Km^R^)	This work
BJa5 pSEVA232_Cel5A	BJa5 carrying plasmid Pseva232-Cel5A (Km^R^)	This work

## Data Availability

Data available upon request to the corresponding author.

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
