# Peer review of "Synthetic Biology Toolkit for a New Species of Pseudomonas Promissory for Electricity Generation in Microbial Fuel Cells"

_microorganisms, 2023, doi:10.3390/microorganisms11082044_

Round 1

Reviewer 1 Report

I would like to thank you for the invitation to review this article of such quality!

All the suggestions deposited here are to try to help with some adjustments.

Line 13 and 14 - the sentence gives an idea that the machinery built is only capable of acting on the two residues mentioned. I suggest modifying it!

Line 22 - I suggest authors substitute the word non for novel.

Line 73 - Biodiesel is not waste, its production and conversion are capable of producing waste. I suggest reviewing this section.

Line 91: I suggest removing the word there.

Line 96: The authors cite the performance of Pseudomonas sp Bja5 in MFCs, but they do not cite any work in which this microorganism has been used for this purpose to justify this statement. I suggest adding this information to the text.

Line 135 to 140: I suggest authors review this part of the manuscript. The authors must correct the acronyms and also the definitions. In the text the authors refer to the Coulombic efficiency as EC, but in the formula it is EC. The definition of CP is also incorrect (Line 139).

Line 147: remove the words for and period

Line 389: replace preposition in with by

Line 352: In Graph 4C, the behavior of the two tested species against different components is clearly evident. The authors discuss these differences very well for almost all compounds, but they do not do so for citrate, which apparently demonstrates a significant difference between the new species and the control species. Therefore, I suggest that the authors discuss this point in more detail.

Line 406: There is a font formatting error. It is necessary that the authors correct the number that is in exponential.

Line 406: The authors do not cite Figure S1 at any time in the text. Apparently, the authors confused figure S1 with figure S2. Thus, it is necessary to make this correction.

Line 451: I suggest authors discuss the results obtained in this section.

Line 470: I suggest that the authors add in the conclusion all the findings of all the tests performed, including the expression of the 428 mCherry reporter gene and the expression of endoglucanase 5A tests.

Reviewer 2 Report

The improvement of MFC efficiency with application of synthetic biology approaches is became very perspective field of research. Presented paper demonstrates interesting results concerning development of the Pseudomonas strain for further engineering for application in the microbial fuel cell (MFC). Authors show that presented strain BJa5 can produce electricity through glycerol oxidation and can be used for further improvement to increase electrical output of MFC. Paper is written in the good scientific language and can be accepted after minor revision. Main comments are:

1.      In introduction section please add brief information concerning already published works about genetic engineering of different strains Pseudomonas;

2.      Please check how written the name of strain. In the line 96 it BJa5 in the line 101 Bja5, the same typos taking place all over the manuscript;

3.      Line 110 – please mention the source of E.coli DH5α strain;

4.      Formula 1 and 2 please check indexes;

5.      Please check number of your strain in the line 244. If I not mistaken, this is mentioned in the paper strain in the GenBank: https://www.ncbi.nlm.nih.gov/nuccore/JALRNG000000000.1

6.      Please add statistics bars at the figure 4 A and B;

Minor editing of English language required
